# COVID-19 Infection and Vaccination Among Healthcare Workers in a South African Academic Hospital: Burden, Risk Factors, and Protective Trends

**DOI:** 10.3390/ijerph22111707

**Published:** 2025-11-12

**Authors:** Xolelwa Ntsham, Vusumuzi Nkosi, Tladi Daniel Ledibane

**Affiliations:** 1Department of Public Health Medicine, Sefako Makgatho Health Sciences University, Ga-Rankuwa 0204, South Africa; 2Department of Environmental Health, Faculty of Health Sciences, Doornfontein Campus, University of Johannesburg, Johannesburg 2094, South Africa; vusi.nkosi@up.ac.za

**Keywords:** COVID-19, healthcare workers, occupational health

## Abstract

Background: Healthcare workers (HCWs) are at occupational risk from COVID-19. Understanding the profile of infected HCWs is important to guide workplace protections. Objectives: To describe the demographic and clinical characteristics of HCWs infected with COVID-19 and to examine factors associated with vaccination status among infected HCWs. Methods: The study design was cross-sectional, using medical records from health workers working in a tertiary teaching facility in Gauteng, South Africa, from 12 May 2021 to 11 May 2022. The data were analysed using univariate and multiple logistic regression analysis. Statistical significance was set at *p* < 0.05. Results: A total of 1235 HCW records were included. The age ranged from 17 to 64 years. The median age was 38 years (IQR: 32–48). Nurses accounted for the largest proportion of cases, followed by healthcare assistants and physicians. Female sex, older age, and comorbidity were associated with higher odds of being vaccinated among infected HCWs. Conclusions: This study provides a descriptive profile of HCWs infected with COVID-19 during the third and fourth wave transition in South Africa. The findings highlight demographic and clinical factors linked to vaccination status among infected HCWs and underscore the continued need for infection prevention and control measures to protect frontline staff.

## 1. Introduction

Healthcare workers (HCWs) are at risk of contracting COVID-19 through their work due to extended exposure to patients with the virus. European studies reported cluster outbreaks among HCWs before implementing universal mask-wearing and other non-pharmaceutical measures [1]. Similarly, cluster outbreaks occurred in China and London, where transmission was limited to community spread, reinforcing the evidence of transmission from SARS-CoV-2-positive patients [2].

In the context of Sub-Saharan Africa, the risk to HCWs remains significant. Studies have shown that HCWs in this region are at a higher risk of infection [3]. The seroprevalence among HCWs ranged from 0% to 45.1%, with factors such as old age, working as a nurse, and lower education levels associated with higher seropositivity rates [4]. HCWs and patients admitted for non-COVID-19 related conditions are particularly vulnerable to infection. HCWs in direct contact with patients are susceptible to transmittable diseases and may play a role in the nosocomial transmission of infectious diseases [5].

Furthermore, factors such as age and overall health play a significant role in COVID-19 outcomes. For instance, younger adults experience better health outcomes following SARS-CoV-2 infection, and children are minimally affected [6]. On the other hand, comorbidities like metabolic disorders and cardiovascular conditions have been identified as risk factors for severe COVID-19 outcomes [7]. Furthermore, a systematic review indicated that conditions such as hypertension, diabetes, respiratory and renal diseases, malignancies, nervous system diseases, and diabetes are linked to higher mortality rates among the elderly [8].

Initially, vaccination initiatives prioritised healthcare workers (HCWs), older people, and individuals with pre-existing conditions [9]. However, vaccine shortages in developing countries necessitated concentrating efforts solely on HCWs [10]. For example, South Africa’s Sisonke vaccination initiative, which commenced on 17 February 2021, aimed to vaccinate all HCWs aged eighteen and above [11]. The Sisonke Initiative was a Phase 3b study initiative using the Ad26.COV2.S COVID-19 vaccine. The Ad26.COV2.S vaccine, a recombinant adenovirus vector encoding the SARS-CoV-2 spike protein, has demonstrated efficacy against COVID-19 in clinical trials [12].

Despite the rapid development of vaccines, most of those recommended demonstrated efficacy and effectiveness, significantly reducing infection rates, severity, hospitalisations, and mortality [13,14] in real-world settings. Meta-analyses show that full vaccination provides 87–89% protection against SARS-CoV-2 infection and over 90% protection against severe outcomes like hospitalisation and death [15,16].

As the pandemic progressed, mutations in the spike protein led to increased transmissibility and the potential for immune evasion after both natural infection and vaccination [17]. For example, the Omicron variant was more transmissible but associated with less severe disease [18]. Omicron possesses more spike protein mutations than other SARS-CoV-2 variants, including six unique mutations in the S2 region [19]. Although the effectiveness of the vaccines against Omicron was reduced, the vaccinated individuals had better protection than their nonvaccinated counterparts [20]. This variable protection of vaccines against Omicron necessitates ongoing surveillance, particularly among at-risk populations, such as healthcare workers.

The study aims were to describe the demographic and clinical characteristics of HCWs infected with COVID-19 at a tertiary teaching hospital in Gauteng Province and to examine demographic and clinical correlates of vaccination status among these cases.

## 2. Materials and Methods

### 2.1. Study Design and Setting

This retrospective cross-sectional record review was conducted among healthcare workers (HCWs) employed at a tertiary academic hospital in Gauteng Province, South Africa. The hospital serves as a major referral centre with approximately 1330 beds and a workforce of about 5000 staff across multiple professional categories, including medical, nursing, allied health, administrative, and support personnel.

The study analysed occupational health records of HCWs with laboratory-confirmed COVID-19 whose infections occurred between 12 May 2021 and 11 May 2022, corresponding to South Africa’s third and fourth COVID-19 waves. Data abstraction and statistical analysis were conducted after ethics approval was granted on 7 October 2021, in accordance with institutional and national research ethics requirements. No data were accessed before approval.

### 2.2. Study Population

The study population comprised HCWs with a laboratory-confirmed diagnosis of COVID-19 who attended the Occupational Health Service (OHS) clinic for post-isolation assessment during the study period. Both permanent and temporary staff were eligible for inclusion.

Inclusion criteria were: (i) laboratory-confirmed COVID-19 infection by polymerase chain reaction (PCR) or antigen test, and (ii) presentation at the OHS clinic following isolation.

Exclusion criteria were: (i) HCWs who tested negative for COVID-19, (ii) those who did not present to the OHS clinic after infection, and (iii) those presenting for reasons other than COVID-19. If multiple infections were recorded for the same HCW, only the first episode during the study period was included.

### 2.3. Data Sources and Variables

Data were extracted from the OHS clinic register and corresponding patient files using a structured case report form (CRF) developed for the study. The CRF captured the following domains:

Sociodemographic characteristics: Age (categorised as 17–34, 35–54, and ≥55 years), sex, and professional cadre (e.g., nurses, physicians, allied health, administrative staff).

Clinical variables: Presence of comorbidities (hypertension, diabetes, cardiovascular disease, asthma, HIV, or other chronic conditions as documented in the file), reported COVID-19 symptoms, hospital admission, and number of days absent from work.

Vaccination status: Receipt of at least one dose of a COVID-19 vaccine, vaccine type (Ad26.COV2.S or others, where recorded), and number of doses received.

Epidemiological variables: wave period (third vs. fourth wave), classified according to the National Institute for Communicable Diseases (NICD) definition of epidemic waves.

All data were de-identified prior to analysis, and each record was assigned a unique study identifier. Records with missing data on key variables (e.g., vaccination status, age, or comorbidity status) were excluded from relevant analyses but retained for descriptive summaries where possible.

### 2.4. Data Management

Data were entered into Epi-Info and exported to Stata version 15 (StataCorp, College Station, TX, USA) for cleaning and analysis. Consistency and logic checks were performed to ensure data quality. Final datasets were stored on password-protected computers accessible only to the research team.

### 2.5. Data Analysis

Descriptive statistics were used to summarise the characteristics of infected healthcare workers. Categorical variables were presented as frequencies and percentages, while continuous variables were summarised using summary statistics.

Logistic regression analysis was then used to examine factors associated with vaccination status among healthcare workers infected with COVID-19. Both crude and adjusted odds ratios (ORs) with 95% confidence intervals (CIs) were calculated. Variables were included in the multivariable model if they showed an association in bivariate analysis (*p* < 0.1) or were identified in the literature as potential confounders. Multicollinearity among predictor variables was assessed using variance inflation factors, and model fit was evaluated with the Hosmer–Lemeshow goodness-of-fit test and pseudo-R^2^ statistics. A two-sided *p*-value of <0.05 was considered statistically significant. These models describe correlates of vaccination within the infected HCW cohort and do not estimate infection risk or vaccine effectiveness.

## 3. Results

### Participant Characteristics

A total of 1235 healthcare worker (HCW) records met the inclusion criteria. The median age was 38 years (IQR: 32–48), with 35.7% aged 17–34 years, 52.5% aged 35–54 years, and 11.8% aged ≥ 55 years. Most were female (82.7%), and 10.5% had documented comorbidities. Infections occurred during the third (44.6%) and fourth (55.4%) COVID-19 waves. The majority (94.3%) took ≤ 30 days of sick leave, only 0.7% required hospital admission, and no deaths were recorded. Table 1 summarises the demographic and clinical characteristics of the study population.

Nurses accounted for the largest proportion of COVID-19 infections (n = 630), followed by support staff such as cleaners, porters, security personnel, caretakers, and food service aids (n = 185) and clerical or administrative staff including clerks, managers, supervisors, directors, human resource officers, data capturers, and secretaries (n = 169). Physicians (medical officers, registrars, consultants, interns, and professors) represented 99 cases. Allied health professionals (physiotherapists, radiographers, dieticians, psychologists, social workers, occupational therapists, clinical technologists, and environmental health practitioners) contributed 62 cases, while pharmacy staff accounted for 23 cases. A small residual “Other” category (n = 64) included staff recorded in occupational health registers without a more specific professional designation. Figure 1 presents the distribution of infections by professional categories.

About 55.4% of infected HCWs were vaccinated. Vaccination status differed significantly by sex: 57.1% of females versus 47.2% of males were vaccinated (*p* = 0.008). Table 2 presents vaccination status by sex.

A higher proportion of vaccinated HCWs were infected during the fourth wave compared to the third wave (65.1% vs. 43.4%, *p* < 0.001). Table 3 shows vaccination status stratified by wave.

Univariate logistic regression showed that males had significantly lower odds of being vaccinated compared with females (Crude OR = 0.67, 95% CI: 0.49–0.90, *p* = 0.008). Older HCWs (≥55 years) had three-fold higher odds of vaccination than those aged 17–34 years (Crude OR = 2.94, 95% CI: 1.97–4.39, *p* < 0.001). The presence of comorbidity was also positively associated with vaccination (Crude OR = 1.48, 95% CI: 1.02–2.16, *p* = 0.041).

After adjustment in the multivariable model, these associations remained significant. Males continued to show lower odds of vaccination (AOR = 0.65, 95% CI: 0.47–0.89, *p* = 0.007). Compared with HCWs aged 17–34 years, those aged 35–54 years (AOR = 1.84, 95% CI: 1.43–2.38, *p* < 0.001) and those aged ≥ 55 years (AOR = 3.28, 95% CI: 2.13–5.04, *p* < 0.001) were significantly more likely to be vaccinated. Comorbidity remained a predictor of vaccination (AOR = 1.21, 95% CI: 1.01–1.98, *p* = 0.043). In addition, infection during the fourth wave was strongly associated with vaccination status (AOR = 2.54, 95% CI: 2.01–3.23, *p* < 0.001). Table 4 summarises the univariate and multivariable regression results.

## 4. Discussion

This study provides a descriptive profile of healthcare workers infected with COVID-19 during the third to fourth wave transition in South Africa, focusing on demographic and occupational patterns among cases and correlates of vaccination status. The findings show that infections occurred across all staff categories, though nurses accounted for the largest proportion of cases, followed by support and clerical or administrative staff. This pattern is consistent with previous research findings reports that identified nurses and patient-facing personnel as having higher occupational exposure to SARS-CoV-2 due to prolonged, close contact with patients and colleagues [21,22,23,24]. However, the presence of infections among non-clinical groups such as cleaners, porters, and administrative staff highlights that transmission risk extended beyond direct patient care, likely reflecting environmental or interpersonal exposure in shared spaces.

The predominance of female HCWs among infected cases mirrors the gender distribution of the healthcare workforce in South Africa and other low- and middle-income settings [3,7]. While this may partly explain their higher representation, it also aligns with international findings that women, especially nurses, comprise the majority of frontline health personnel, and thus, face greater exposure to respiratory pathogens [22,23].

Although vaccination does not eliminate infection, understanding vaccination patterns among infected HCWs provides valuable insight into health system dynamics during the transition from Delta to Omicron. In this study, vaccination among infected HCWs was more common in older individuals, females, and those with comorbidities. These trends likely reflect perceived susceptibility and prioritisation during the national rollout, rather than vaccine effectiveness per se. Similar demographic correlates of vaccine uptake have been documented in other settings, where older and comorbid individuals demonstrated higher acceptance due to increased perceived risk [25]. Conversely, lower uptake among younger and male HCWs may reflect elements of complacency or vaccine hesitancy, as reported in several international studies. For example, reliance on non-medical or social media sources of information has been linked to vaccine refusal intentions, while gender and the source of health information have been shown to significantly influence hesitancy patterns [26,27]. These findings reinforce the importance of context-sensitive communication strategies and transparent information dissemination to build trust and improve vaccine confidence among healthcare workers.

The temporal distribution of infections across waves further illustrates the changing pandemic dynamics. A larger number of cases occurred during the fourth (Omicron-driven) wave despite higher vaccination coverage, consistent with Omicron’s high transmissibility and partial immune evasion. These observations are descriptive and should not be interpreted as evidence of vaccine failure or protection, as the study lacked denominators, vaccination dates, and severity outcomes to assess effectiveness.

Overall, this study highlights the diverse occupational impact of COVID-19 across the health workforce and underscores the continuing need for robust infection prevention and control practices. Targeted interventions should prioritise high-contact professionals such as nurses, as well as support and administrative staff who may be overlooked in IPC programmes. Ongoing staff education, adequate provision of personal protective equipment, and accessible vaccination services are essential to safeguard healthcare workers and ensure continuity of service delivery during future outbreaks.

### Implications

The findings of this study highlight the need for sustained system-wide measures to protect healthcare workers across all professional categories. Nurses, support staff, and administrative personnel together accounted for most infections, highlighting that occupational exposure extends beyond direct patient care. Infection-prevention and control strategies should therefore target all professional categories, including non-clinical workers who may have limited access to training and protective equipment.

Vaccination coverage among infected HCWs was higher among older, female, and comorbid staff, suggesting that age, gender, and perceived vulnerability influence vaccine uptake. Communication and outreach strategies should address the specific concerns and motivational drivers of younger and male HCWs, who demonstrated lower uptake. Evidence from other settings emphasises the role of trusted information sources in improving vaccine confidence and countering misinformation.

Health institutions should maintain continuous IPC education, ensure adequate PPE availability, and integrate vaccination promotion into occupational health programmes. Collaborative efforts between management, occupational health teams, and frontline staff can strengthen preparedness and resilience for future infectious disease outbreaks.

## 5. Limitations

The following limitations are acknowledged. First, only COVID-19-positive employees who presented at the occupational health clinic were included in the study. The absence of a denominator population (i.e., uninfected HCWs or person-time at risk) means that we could not estimate infection prevalence, incidence, or relative risks across professional categories, sexes, vaccination groups, or waves. The findings therefore reflect the composition of cases only and should not be interpreted as measures of infection risk. Second, vaccination dates, completeness of vaccination schedules, and booster status were not consistently recorded in occupational health records. As a result, we could not define breakthrough infections (e.g., ≥14 days post-vaccination) or assess time-since-vaccination effects. All descriptions of vaccination patterns should be interpreted as correlates within the infected cohort, not as evidence of vaccine effectiveness.

Third, poor record quality and missing information may have introduced misclassification or reduced the accuracy of reported comorbidities and vaccination status.

Fourth, we could not distinguish between occupationally acquired versus community-acquired infections, as detailed exposure assessments were not routinely documented.

Finally, the findings may not be generalisable to all South African HCWs, as the study was restricted to one academic hospital and included only those who reported to the occupational health clinic.

## 6. Conclusions

This study describes the characteristics of healthcare workers infected with COVID-19 at a South African tertiary hospital during the Delta–Omicron transition. Nurses accounted for most infections, followed by support and clerical staff, indicating exposure risk across multiple workforce categories.

Vaccination among infected HCWs was more common in older, female, and comorbid staff, reflecting demographic patterns of vaccine uptake. These observations are descriptive and do not imply differences in infection risk or disease severity.

Strengthening infection-prevention measures and promoting vaccine uptake among younger and male HCWs remain essential to protect the healthcare workforce in future outbreaks.

## Figures and Tables

**Figure 1 ijerph-22-01707-f001:**
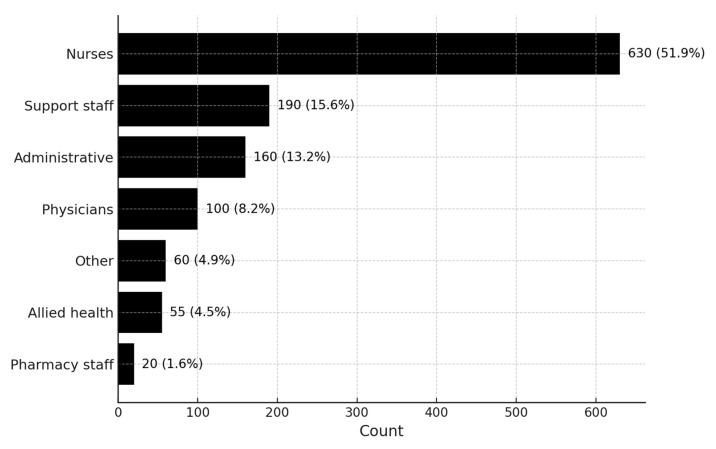
COVID-19 infections among healthcare workers per professional category.

**Table 1 ijerph-22-01707-t001:** Characteristics of healthcare workers infected with COVID-19 (n = 1235).

Characteristic	Frequency (n)	Percentage (%)
Sex		
Female	1021	82.7
Male	214	17.3
Age category (years)		
17–34	440	35.7
35–54	647	52.5
≥55	146	11.8
Comorbidity		
No	1105	89.5
Yes	130	10.5
Wave period		
Third wave	551	44.6
Fourth wave	684	55.4
Days of absence		
1–14	569	46.1
15–30	595	48.2
>30	71	5.8
Outcome		
Not admitted	1227	99.3
Admitted	8	0.7

**Table 2 ijerph-22-01707-t002:** Vaccination status among healthcare workers infected with COVID-19, stratified by sex (n = 1235).

Sex	Vaccinated		Not Vaccinated		Total		*p*-Value
	n	%	n	%	n	%	
Female	583	57.1	438	42.9	1021	82.7	0.008
Male	101	47.2	113	52.8	214	17.3	
Total	684	55.4	551	44.6	1235	100.0	

**Table 3 ijerph-22-01707-t003:** Vaccination status among healthcare workers infected with COVID-19, stratified by wave period (n = 1235).

Wave Period	Vaccinated		Not Vaccinated		Total		*p*-Value
	n	%	n	%	n	%	
Third wave	239	43.4	312	56.6	551	44.6	<0.001
Fourth wave	445	65.1	239	34.9	684	55.4	
Total	684	55.4	551	44.6	1235	100.0	

**Table 4 ijerph-22-01707-t004:** Factors associated with vaccination among healthcare workers infected with COVID-19. Logistic regression analysis (n = 1235).

Characteristic	Crude OR (95% CI)	*p*-Value	Adjusted OR (95% CI)	*p*-Value
Sex				
Female (ref)	1.00	–	1.00	–
Male	0.67 (0.49–0.90)	0.008	0.65 (0.47–0.89)	0.007
Age (years)				
17–34 (ref)	1.00	–	1.00	–
35–54	1.89 (1.48–2.41)	<0.001	1.84 (1.43–2.38)	<0.001
≥55	2.94 (1.97–4.39)	<0.001	3.28 (2.13–5.04)	<0.001
Comorbidity				
No (ref)	1.00	–	1.00	–
Yes	1.48 (1.02–2.16)	0.041	1.21 (1.01–1.98)	0.043
Wave period				
Third wave (ref)	1.00	–	1.00	–
Fourth wave	2.43 (1.93–3.06)	<0.001	2.54 (2.01–3.23)	<0.001

Abbreviations: OR = odds ratio; CI = confidence interval; ref = reference category.

## Data Availability

The data are not publicly available due to confidentiality of hospital staff records but may be made available from the corresponding author upon reasonable request.

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
