# Peer review of "COVID-19 Infection and Vaccination Among Healthcare Workers in a South African Academic Hospital: Burden, Risk Factors, and Protective Trends"

_ijerph, 2025, doi:10.3390/ijerph22111707_

Round 1

Reviewer 1 Report

Comments and Suggestions for Authors

The manuscript is of interest to specialists and decision makers, in the context of possible emerging infectious diseases in the field of public health, and in the field of occupational health.
The manuscript requires minor corrections, as follows:
1. Abstract, lines 19-20, to verify “The age of the participants varied from 1 to 64 years, with 19 an average age of 39.6 years.” To be corrected.
2. Line 58: the elderly[8]8. To be corrected.
In the Data Analysis section it is mentioned “... medians with interquartile ranges (IQR),…”. Where in the text of the manuscript are they used?
3. Lines 149-151, the phrase: “This section may be divided by subheadings. It should provide a concise and precise description of the experimental results, their interpretation, as well as the experimental conclusions that can be drawn.” What is it? It does not belong here.
4. Row 154: mean age was 39.6 years (SD ± [insert if available]). To be corrected.

Author Response

I have attached a table detailing the corrections 

Reviewer 2 Report

Comments and Suggestions for Authors

This retrospective cross‑sectional record review describes 1,235 healthcare workers (HCWs) with laboratory‑confirmed COVID‑19 who presented to an occupational health service at a tertiary hospital in Gauteng between 12 May 2021 and 11 May 2022. The study summarises demographics, comorbidities, wave period, days of absence and hospitalisation, and uses logistic regression to examine factors associated with being vaccinated among infected HCWs. Most cases were female (82.7%), mean age 39.6 years, with very low hospitalisation (0.7%). Vaccinated individuals constituted 55.4% of infected HCWs, with a higher vaccinated share among cases in the fourth wave versus the third. Older age, comorbidity, and female sex were associated with higher odds of being vaccinated among those infected.

The topic is relevant and the dataset spans the Delta–Omicron transition in South Africa, a period of clear policy interest. However, the current case‑only design and several reporting inconsistencies limit the strength of inference. Claims about “risk factors for infection” and protective trends are not supported without denominators or person‑time; the regression models explain vaccination among cases, not infection risk. Substantial reframing and corrections are required before the paper can meet the standards of a high‑impact journal. I recommend major revision focused on aligning aims with design, fixing reporting issues, and sharpening the interpretation.

Major comments. First, there seems to be a fundamental aims–design mismatch. The aims include estimating point prevalence and “risk factors associated with COVID‑19 infection” (lines 81–85), yet the sample consists only of HCWs who were infected and presented for post‑isolation assessment (lines 93–104). Without a denominator of uninfected HCWs or person‑time at risk, neither prevalence nor infection risk by cadre, sex, vaccination, or wave can be estimated. Please either obtain appropriate denominators (e.g., headcounts by cadre and vaccination coverage by wave) and analyse rates, or reframe the paper as a descriptive profile of infected HCWs, avoiding causal language about infection risk. The bar chart on page 5 further underscores this issue by mixing professional categories with “direct/indirect contact” groupings without denominators, which can only describe the composition of cases, not relative risk.

Second, clarify what the regression estimates and temper the causal interpretation. Table 4 models the odds of being vaccinated among infected HCWs, not the odds of infection. This is a study of vaccination correlates within cases; it cannot speak to vaccine effectiveness or protection against infection without denominators, vaccination dates, and time at risk. Statements in the Abstract that vaccinated HCWs had fewer severe infections in the third wave and interpretations that imply protection should be reframed as descriptive observations about case composition unless supported by analyses with appropriate denominators and severity outcomes by vaccination status and wave. Consider, instead, modelling clinically relevant outcomes among cases (e.g., hospitalisation or prolonged absence) as the dependent variable.

Third, define vaccination exposure precisely and address timing. The Methods and Limitations acknowledge missing vaccination dates (lines 145–147, 269–276), but the manuscript discusses breakthrough infections and wave‑specific patterns. Please define “vaccinated” (e.g., ≥1 dose vs full schedule; vaccine type; booster status), specify a breakthrough definition (e.g., ≥14 days post‑dose), and incorporate time‑since‑vaccination where available. If these data cannot be recovered, claims about breakthrough infection, protection by wave, or attenuation of severity should be softened accordingly.

Fourth, I suggest resolving reporting inconsistencies and present percentages transparently. In the text, “57.1% of females versus 47.2% of males were vaccinated” is asserted, but Table 2 appears to report overall percentages by sex rather than row percentages within sex; as a result, the text contradicts the table (lines 176–180). Choose a single approach—preferably row percentages for stratified tables—and ensure text and tables align. Similarly, the Abstract reports an age range of 1–64 years for a HCW cohort, which is implausible and likely erroneous (line 19). Figure 1 on page 5 should use mutually exclusive categories with clear legends for the asterisks and, if risk comparisons are intended, cadre‑specific denominators. Finally, consider situating the finding that vaccination among cases was more common in older, comorbid, and female HCWs within the vaccine‑hesitancy literature and the role of information sources (Del Riccio, 2022; Sieber, 2022), which would enrich the policy implications.

Minor comments: Line 11: “at risk for occupational health from COVID‑19 infection” → “at occupational risk from COVID‑19 infection.” Line 19: age range “1 to 64 years” is implausible for HCWs; verify and correct. Line 21: “Older healthcare workers individuals” contains a duplication. Lines 123–127 and 142–147: duplicated “Data Management” sections; please consolidate into one. Lines 149–151: residual template text in Results should be removed. Lines 153–159 and again 160–167: the paragraph beginning “A total of 1,235…” is duplicated; retain a single instance. Lines 176–180: the sentence on vaccination prevalence by sex conflicts with Table 2 and “table 2” should be capitalised; standardise to row percentages and fix the text–table mismatch. Page 5 figure: add a legend explaining the asterisks, avoid mixing cadre with contact‑type categories, and add denominators if risk comparisons are intended. Line 298: “funding,.” has an extra punctuation mark.

Refs:

Sieber WJ, Achar S, Achar J, Dhamija A, Tai-Seale M, Strong D. COVID-19 vaccine hesitancy: Associations with gender, race, and source of health information. Fam Syst Health. 2022;40(2):252-261. doi:10.1037/fsh0000693

Del Riccio M, Bechini A, Buscemi P, Bonanni P, On Behalf Of The Working Group Dhs, Boccalini S. Reasons for the Intention to Refuse COVID-19 Vaccination and Their Association with Preferred Sources of Information in a Nationwide, Population-Based Sample in Italy, before COVID-19 Vaccines Roll Out. Vaccines (Basel). 2022;10(6):913. Published 2022 Jun 8. doi:10.3390/vaccines10060913

Author Response

I have attached the responses on the attached table 

Round 2

Reviewer 2 Report

Comments and Suggestions for Authors

I thank the authors for implementing my comments.